# Current Status of Staphylococcal Cassette Chromosome *mec* (SCC*mec*)

**DOI:** 10.3390/antibiotics11010086

**Published:** 2022-01-11

**Authors:** Yuki Uehara

**Affiliations:** 1Department of Microbiology, Faculty of Medicine, Juntendo University, Tokyo 113-0033, Japan; yuuehara@juntendo.ac.jp or yukiue@luke.ac.jp; Tel.: +81-3-3541-5151; 2Department of General Medicine, Faculty of Medicine, Juntendo University, Tokyo 113-0033, Japan; 3Department of Clinical Laboratory, St. Luke’s International Hospital, Tokyo 104-8560, Japan; 4Department of Infectious Diseases, St. Luke’s International Hospital, Tokyo 104-8560, Japan

**Keywords:** methicillin-resistant *Staphylococcus aureus*, staphylococcal cassette chromosome *mec* (SCC*mec*), SCC*mec* typing, SCC*mec* subtyping, International Working Group on the Classification of Staphylococcal Cassette Chromosome Elements

## Abstract

Staphylococcal cassette chromosome *mec* (SCC*mec*) typing was established in the 2000s and has been employed as a tool for the molecular epidemiology of methicillin-resistant *Staphylococcus aureus*, as well as the evolution investigation of *Staphylococcus* species. Molecular cloning and the conventional sequencing of SCC*mec* have been adopted to verify the presence and structure of a novel SCC*mec* type, while convenient PCR-based SCC*mec* identification methods have been used in practical settings for many years. In addition, whole-genome sequencing has been widely used, and various SCC*mec* and similar structures have been recently identified in various species. The current status of the SCC*mec* types, SCC*mec* subtypes, rules for nomenclature, and multiple methods for identifying SCC*mec* types and subtypes were summarized in this review, according to the perspective of the International Working Group on the Classification of Staphylococcal Cassette Chromosome Elements.

## 1. Introduction

Methicillin, a semisynthetic penicillin that specifically targets beta-lactamase-producing staphylococci, was introduced to the medical field in 1960. However, in the same year (1960), methicillin-resistant *Staphylococcus aureus* (MRSA) was identified [1,2]. Since then, MRSA has been the most prevalent and well-known antimicrobial-resistant bacteria for more than 60 years. In the first report of MRSA, it was described that MRSA existed before starting the use of methicillin but was rarely isolated in clinical settings, but after that, the prevalence of MRSA in healthcare settings increased gradually. For example, in the UK, the proportion of isolates of *S. aureus* from blood cultures that were methicillin-resistant increased in the 1990s and reached approximately 40% around 2000, though it decreased to less than 10% in the 2010s [3,4]. A high prevalence of antimicrobial resistance not only to methicillin but also to other antimicrobials was thought to have correlations with the inappropriate use of antimicrobials, so the prevalence of MRSA is still used as an indicator of good infection control and prevention practices and appropriateness of antimicrobial usage [5,6].

MRSA is known to produce an additional penicillin-binding protein designated as PBP2’ (PBP2a). PBP2’ exhibits a low affinity to most semisynthetic penicillin, such as methicillin, nafcillin, and oxacillin, as well as most cephalosporins. With its low affinity to beta-lactams and encoding gene *mecA*, PBP2’ was first reported in the mid-1980s [7,8,9]. Later, it was revealed that *mecA* is surrounded by genes that control its expression, *mecR1* (encoding the signal transducer protein mecR1), and *mecI* (encoding the repressor protein mecI) [10,11]. When *mecA* was determined to be widely disseminated among multiple staphylococcal species, it was hypothesized that *mecA* could be carried on a mobile genetic element that could be transferred among staphylococcal species (Figure 1).

Mobile genetic elements containing *mecA* and their regulatory genes were designated as staphylococcal cassette chromosome *mec* (SCC*mec*) and were first reported in strain N315 (Sequence type 5, NY/JAPAN clone), of which the whole-genome sequence was subsequently revealed, followed by other strains, NCTC10442 and 85/2082 [12,13,14]. The SCC*mec* in NCTC10442, N315, and 85/2082 were designated as SCC*mec* I, SCC*mec* II, and SCC*mec* III, respectively [13].

The precise excision of SCC*mec* and integration of plasmid containing *ccrA* and *ccrB* was proven by experiments within a 24-h incubation period using N315 having SCC*mec* II [12]. However, the excision and integration of SCC*mec* from/to the *S. aureus* isolate is thought to happen in a complexed process in real clinical settings. It was observed that the excision and integration of small SCC*mec* such as SCC*mec* I could happen, but the integration of SCC*mec* II and V, which were larger than SCC*mec* I (>45 kb), could not happen even in vitro [15]. For large SCC*mec* structures, additional processes may be necessary to integrate into *S. aureus*, such as conjugative plasmids as a carrier or subsequent multiple recombination events. As an example, it was reported that the two isolates from a patient with chronic *S. aureus* infection of an intracardiac device changed from MSSA to MRSA during antimicrobial treatment, and the latter isolate possessed SCC*mec* II [16]. Some inversions and extra genetic fragments, which were not found in the original SCC*mec* II and its surrounding regions of N315, suggested that SCC*mec* II in the latter isolate did not directly integrate from N315, and additional processes might happen for integration.

## 2. International Working Group on the Classification of Staphylococcal Cassette Chromosome Elements (IWG-SCC)

Since the first reports of SCC*mec* I, II, and III in the early 2000s, various SCC*mec* elements have been reported by different researchers worldwide, and in addition to being adopted as a molecular epidemiology tool in healthcare settings, these elements have been utilized in the research on the evolution of staphylococci. These studies have also raised concerns about the confusion of the SCC*mec* nomenclature and the loss of its value. Therefore, the IWG-SCC released a report on the “Classification of Staphylococcal Cassette Chromosome *mec* (SCC*mec*): Guidelines for Reporting Novel SCC*mec* Elements” in 2009, followed by the Guidelines for Reporting Novel *mecA* Gene Homologues [17,18].

Recently, the IWG-SCC is updating its website, which contains information about IWG-SCC members, curators, requesting representative isolates of SCC*mec* types, current SCC*mec* types and subtypes approved by IWG-SCC, and IWG-SCC recommendations for designating and reporting new SCC*mec* elements (https://www.sccmec.org, under revision, accessed on: 14 December 2021).

## 3. Structure of SCC*mec*

The description provided in this section is based on the IWG-SCC recommendations, and additional information after the publication of these recommendations is presented [17,18].

### 3.1. Requirement for Defining SCCmec

According to the IWG-SCC recommendations, SCC*mec* is mainly characterized by four factors: (i) carriage of *mecA* in a *mec* gene complex, (ii) carriage of a *ccr* gene(s) (*ccrAB* and/or *ccrC*) in the *ccr* gene complex, (iii) integration at a specific site in the staphylococcal chromosome, designated as the integration site sequence (ISS) for SCC, which serves as a target for *ccr*-mediated recombination, and (iv) the presence of flanking direct repeat sequences containing the ISS [17]. As an example, a comparison of the structures of SCC*mec* I–V is presented in Figure 2. Several homologs of *mecA* have been reported recently, and instead of *mec*A, *mecC* is present in SCC*mec* IX. Details of the *mecA* homologs are described later in this review.

### 3.2. ccr Gene Complex

The *ccr* gene complex comprises the *ccr* gene(s) and surrounding open reading frames, several of which have unknown functions. Currently, three phylogenetically distinct *ccr* genes, *ccrA*, *ccrB*, and *ccrC*, have been identified in *S. aureus* with DNA sequence similarities of less than 50%. Although the *ccr* sequences present in staphylococci other than *S. aureus* are remarkably diverse, novel *ccr* genes should be defined based on DNA sequence similarities of <50%, while novel allotypes of *ccr* genes should be designated if their DNA sequence similarity identities are between 50% and 85% [17]. Figure 3 presents a summary of the *ccrA*, *ccrB*, and *ccrC* genes and their allotypes. Wide implementations of whole-genome sequencing in the field of bacteriology may identify more diverse *ccr* genes from multiple species other than staphylococci; however, presently, this convention should be followed when naming novel *ccr* genes.

The *ccr* gene complexes were numbered according to the timing of their descriptions. To date, two distinct groups have been reported: one carrying two adjacent *ccr* genes (*ccrA* and *ccrB*) and the other carrying only *ccrC*. For example, the *ccr* gene complexes identified in *S. aureus* include type 1 (carrying *ccrA1B1*), type 2 (carrying *ccrA2B2*), type 3 (carrying *ccrA3B3*), type 4 (carrying *ccrA4B4*), and type 5 (carrying *ccrC*).

### 3.3. mec Gene Complex

The *mec* gene complex comprises *mecA*, its regulatory genes, and the associated insertion sequences. Currently, three classes of the *mec* gene complex are recognized [17]. The class A *mec* gene complex (class A *mec*) is a prototype contained in SCC*mec* II. The class A *mec* gene complex contains *mecA*, the complete *mecR1* and *mecI* regulatory genes upstream of *mecA*, and the hypervariable region (HVR) and insertion sequence IS*431* downstream of *mecA*. The class B *mec* gene complex (class B *mec*) comprises *mecA*, a truncated *mecR1* resulting from the insertion of IS*1272* upstream of *mecA*, and HVR and IS*431* downstream of *mecA*. The class C *mec* gene complex (class C *mec*) contains *mecA* and truncated *mecR1* via the insertion of IS*431* upstream of *mecA* and the HVR and IS*431* downstream of *mecA*.

In the 21st century, multiple *mecA* homologs have been identified from staphylococci other than *S. aureus* and *Macrococcus caseolyticus*. Hence, the IWG-SCC released a commentary on the “Guidelines for Reporting Novel *mecA* Gene Homologues” in 2012 (Table 1) [18].

The *mecA* genes with nucleotide sequence identities similar to the original *mecA* gene by 95% are designated as *mecA*, thus indicating that they are members of the allotype represented by the original *mecA* gene found in *S. aureus* N315. Those with nucleotide sequence identities to the original *mecA* of 70–95% are regarded as belonging to other allotypes of *mecA*. Accordingly, the *mecA* homologs detected in *Staphylococcus sciuri*, with a nucleotide sequence identity to the original *mecA* gene of approximately 80%, are designated as *mecA1*. The *mecA* homologs in *Staphylococcus vitulinus,* with nucleotide sequence identities similar to the original *mecA* gene of approximately 90%, are designated as *mecA2*.

In addition, the *mec* gene types are divided into allotypes, where each allotype encompasses a group of *mec* genes that share 70–95% nucleotide sequence identities to *mecA* of *S. aureus* N315 and *mecB* of *Macrococcus caseolyticus* JCSC5402, which was originally reported as *mecAm*, or *mecC* of *S. aureus* LGA251, which was isolated from bulk milk, daily cattle, and humans and originally designated as *mecA*_LGA251_ [19,20]. Again, wide implementation of whole-genome sequencing may identify more diverse *mec* genes from multiple species other than staphylococci; however, presently, this convention should be followed when naming novel *mec* genes.

### 3.4. J Regions

In addition to the *mec* and *ccr* gene complexes, the SCC*mec* element also contains three J regions (previously called “junkyard regions”), which constitute nonessential components of the chromosome cassette. However, J regions may carry additional antimicrobial resistance determinants, and variations in the J regions within the same *mec-ccr* complex are adopted to define SCC*mec* subtypes. As illustrated in Figure 2, the J1 region (formerly called L-C) is located between the right chromosomal junction and the *ccr* complex, J2 (formerly called C-M) is between the *ccr* gene complex and the *mec* gene complex, and J3 (formerly called I-R) is between the *mec* complex and the left chromosomal junction.

## 4. Nomenclature of SCC*mec* and Its Subtypes

SCC*mec* elements are classified into types and subtypes by hierarchical systems. Currently, two methods are used to describe SCC*mec* types (Figure 4). These SSCC*mec* types are described by the combination of the *mec* gene complex class and the *ccr* gene complex type. SCC*mec* types can also be described using Roman numerals, such as SCC*mec* types I, II, and III. Generally, these two descriptions are reported and written together, for example, SCC*mec* type I (1B), SCC*mec* type II (2A), and SCC*mec* type III (3A). 

SCC*mec* subtypes can be classified by differences in the J regions. The J regions contain characteristic genes, pseudogenes, noncoding regions, and mobile genetic elements such as insertion sequences and plasmids or transposons, which are used to define SCC*mec* subtypes. In the 2000s, some SCC*mec* subtypes could be reported with additional sequencing results of the J regions; however, this approach has a potential risk for subtype misclassification. Therefore, it is necessary to identify and report new SCC*mec* subtypes based on the entire nucleotide sequence. The following three methods have been adopted to describe subtypes of SCC*mec*: (i) expressing the J region differences as small letters, e.g., IVa, IVb, and IVc; (ii) expressing the differences due to the presence or absence of mobile genetic elements as capital letters, e.g., IA, IIA, and IVA; and (iii) expressing the differences in each J1, J2, and J3 region in Arabic numbers, which are designated according to the order of discovery, for example, II.1.1.1, II.1.1.2, and II.2.1.1. Currently, expressing J region differences as small letters is the most widely used to describe SCC*mec* subtypes. J1 region differences are mainly used to differentiate subtypes; however, some subtypes are characterized by unique contents in the J3 region.

## 5. Current SCC*mec* Types and Subtypes

The latest list of SCC*mec* types is presented in Table 2 and Figure 5, with the first official reports as references. Table 2 describes the details of the current SCC*mec* types, including the combinations of the *ccr* complex and *mec* complex and representative strains and their source, country, and reported year. Figure 5 shows the comparison of the structures of SCC*mec* listed in Table 2. To date, 14 SCC*mec* types have been officially reported and approved by the IWG-SCC.

SCC*mec* subtypes I, II, IV, and V are also presented in Table 3, together with their references. Table 3 shows the J1 region-based subtypes described by adding small letters, with one exception in SCC*mec* subtype IVn, which differs from the SCC*mec* subtype IVc in the J3 region. In addition to the J1 region-based subtypes, SCC*mec* subtypes IIB, IVE, and IVF were designated using a different method to express the differences due to the presence or absence of mobile genetic elements in the entire SCC*mec* structure as capital letters [30].

## 6. Methods to Identify SCC*mec* Types and Subtypes

### 6.1. PCR-Based Methods

PCR-based methods have been widely used to detect important components to differentiate SCC*mec* types and subtypes because of their relatively easy access to reagents and equipment. Single PCR and multiplex PCR are used to identify SCC*mec* types. Single PCR for each genomic component in SCC*mec* is a basic, generally sensitive, and specific method. The SCC*mec* type is identified by the combination of the detected components. However, it is difficult to guess the SCC*mec* type before analysis, and numerous different single PCRs are necessary for identification of the SCC*mec* type. Therefore, multiplex PCR methods have been reported as useful tools to detect multiple components in SCC*mec* and identify the SCC*mec* types and subtypes efficiently in one analysis, even though the sensitivity and specificity to detect each component were sometimes less than single PCRs. The multiplex PCR methods to which the IWG-SCC members contributed and were widely used are introduced here in this review. Dr. Zhang reported a multiplex PCR assay for characterization and concomitant subtyping of SCC*mec* types I, II, III, VIa, b, c, d, and V in 2005 and its updated version in 2012 [43,44]. Dr. Kondo formulated a combination of multiplex PCRs for rapid SCC*mec* type assignments, which could differentiate SCC*mec* types I, II, III, IV, V, VI, and representative J regions to define subtypes in 2007 [34]. In 2007, Dr. Milheirico communicated multiplex PCR strategies for the assignment of SCC*mec* types I, II, III, IV and its major subtypes, and V [37,45].

The limitations of multiplex PCR methods should be considered when interpreting the results. For example, the partial deletion of the *ccrB2* gene present in Japanese USA300, which was designated as ψUSA300, could cause failure in identifying the SCC*mec* type in some multiplex PCR methods [46]. In 2020, Dr. Yamaguchi published a substantially beneficial review covering the history and concept of SCC*mec* and detailed the methods of SCC*mec* typing using multiplex PCR formulated by Dr. Kondo [34,47]. The limitations of multiplex PCR methods, such as problems in identifying the SCC*mec* type if one isolate had both SCC*mec* and SCC (i.e., structure of SCC*mec* without the *mec* complex) components or the failure of PCR caused by the deletion/insertion of sequences of targeted sites were discussed. Approaches for solutions of the problems were also mentioned, mainly about the necessity of additional single PCRs to detect indefinite components in SCC*mec* with different primer sets and PCR conditions.

### 6.2. Whole-Genome Sequencing-Based Methods

SCC*mec*Finder is a web-based tool for SCC*mec* typing using whole-genome sequences and was released by Dr. Kaya et al. in 2018 (https://cge.cbs.dtu.dk/services/SCCmecFinder/, accessed on 14 December 2021). SCC*mec*Finder identifies SCC*mec* elements in sequenced *S. aureus* isolates. The read data from major platforms for whole-genome sequencing or preassembled genome/contigs can be submitted to the website of SCC*mec*Finder. Users can obtain information about the prediction of SCC*mec* types based on the genes in the *ccr* complex, *mec* complex, and J regions, including the homology to the entire cassette [48]. Recently, a rapid method to detect MRSA, covering multiple SCC*mec* types and subtypes, directly from clinical samples has been developed [49]. The rapid confirmation of MRSA followed by whole-genome sequencing using the same samples and implementation of SCC*mec*Finder may contribute to reveal the diversity of SCC*mec* in various settings. However, as mentioned later, the objective confirmation of the structure is necessary to report new SCC*mec,* even though whole-genome sequencing has the potential to discover new ones easily.

## 7. How to Report New SCC*mec* and SCC*mec* Subtypes in the Era of Whole-Genome Sequencing: Role of IWG-SCC

Molecular cloning and conventional sequencing of SCC*mec* have been adopted for several years to identify and verify the structures of new SCC*mec*. However, these methods have rarely appeared in recent publications regarding new SCC*mec* elements. Alternatively, whole-genome sequencing has been widely used. In the era of whole-genome sequencing, in addition to *S. aureus*, numerous *mec* homologs, *ccr* homologs, and other components of SCC*mec* can be identified from other staphylococci and numerous species other than staphylococci. Furthermore, new SCC*mec* elements can be identified more frequently in isolates from animals or the environment. To maintain the value of identification of the SCC*mec* elements as a tool for molecular epidemiology, investigations on the evolution of bacteria and the relationship with antimicrobial resistance, researchers should contact the curator and obtain approval from the IWG-SCC before reporting/publishing new SCC*mec* and/or SCC*mec* subtypes to avoid duplicated names/numbers and reporting structures that are not SCC*mec*.

As aforementioned, a complete SCC*mec* sequence is necessary. The IWG-SCC strongly recommends the use of long-read sequencing technology (e.g., nanopore systems, PacBio systems, etc.) to verify the sequence of a new cassette chromosome, because the assembly of the entire SCC*mec* sequence sometimes solely fails with the results of short-read sequencing technology, possibly due to multiple insertion sequences.

To maintain the appropriate nomenclature of SCC*mec* types and subtypes, researchers are requested to send the isolates containing candidates of new SCC*mec* elements to the Statens Serum Institut (SSI) in Denmark and/or the National Institute of Infectious Diseases (NIID) in Japan, both of which serve as reference laboratories and reference strain banks of SCC*mec*. Subsequently, SSI and NIID will conduct long-read whole-genome sequencing and annotation to verify the suggested nomenclature by researchers. After depositing the strains to SSI and NIID, SSI and NIID will share the strains when requested from outside to contribute to SCC*mec* research worldwide. The original researchers will be acknowledged and asked permission to share the strains from SSI and/or NIID whenever necessary.

To date, complete structures of SCC*mec* present in *S. aureus* have been designated as new SCC*mec* type names, regardless of the host. However, the IWG-SCC has decided not to annotate new SCC*mec* subtypes in species other than *S. aureus* owing to the high complexity of the elements present in isolates other than *S. aureus*. An alternative nomenclature is “SCC*mec*_[NAME OF THE STRAIN]_”, which has already been adopted for non-aureus staphylococci. Discussions should be continued on the appropriate way of differentiating and naming the SCC*mec* structures present in both *S. aureus* and other species using whole-genome sequencing data and/or in silico investigations from public genome databases.

## 8. Conclusions

The discovery of SCC*mec* was based on the findings from multiple researchers about MRSA in the 20th century, and its importance in the fields of molecular epidemiology, infection control and prevention, and investigation of the evolution of MRSA has been developed along with the evolution of molecular analysis technologies. The implementation of whole-genome sequencing will contribute to reveal the entire relatedness of the SCC*mec* elements found not only in MRSA but also in other *Staphylococcus* species and Gram-positive bacteria. Importance as a tool of molecular typing might be declined if whole-genome sequencing is more widely used in clinical settings, but PCR-based methods will continue to be used as easy and costless tools for molecular epidemiology, infection control, and prevention. Therefore, the nomenclature rule of SCC*mec* should be followed, and the IWG-SCC should continue to update the most appropriate policy to designate SCC*mec* elements, especially for isolates other than *S. aureus*, or MRSA isolated from animals or the environment.

## Figures and Tables

**Figure 1 antibiotics-11-00086-f001:**
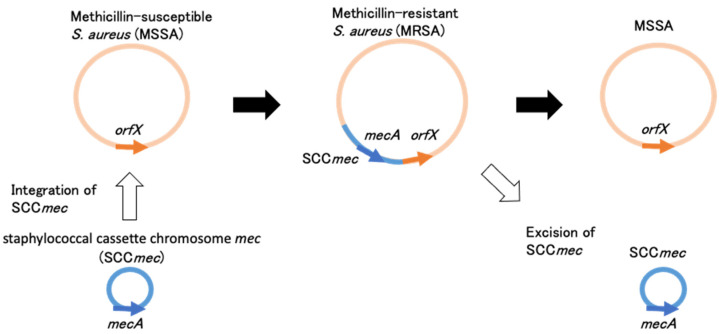
Schematic representation of SCC*mec* excision and integration in *S. aureus*.

**Figure 2 antibiotics-11-00086-f002:**
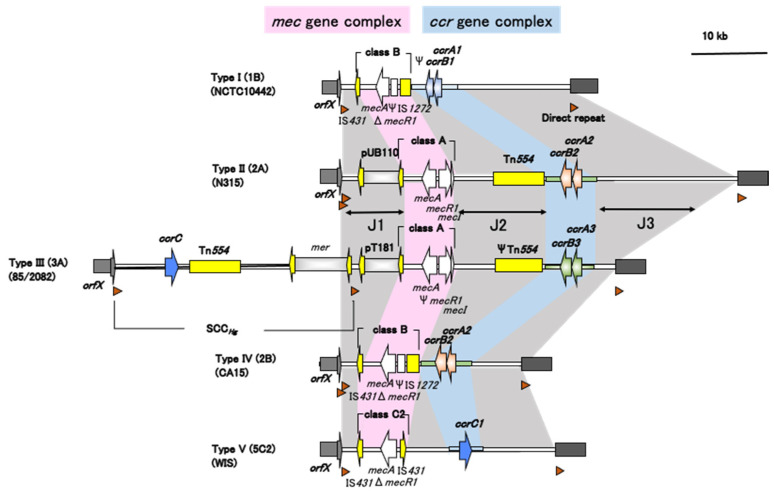
Comparison of the structures of SCC*mec* I–V in *S. aureus*. SCC*mec* should contain the *mec* gene complex and *ccr* gene (complex), with integration site sequence and direct repeats at both ends. Adapted with permission from Ref. [17]. 2009, American Society for Microbiology.

**Figure 3 antibiotics-11-00086-f003:**
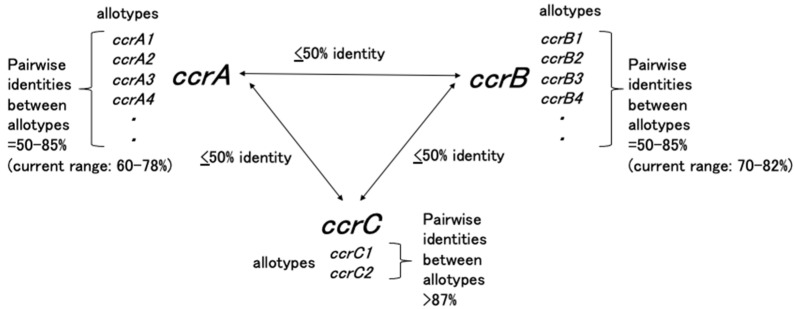
Representation of the naming conventions for *ccr* genes in *S. aureus.* Adapted with permission from Ref. [17]. 2009, American Society for Microbiology.

**Figure 4 antibiotics-11-00086-f004:**
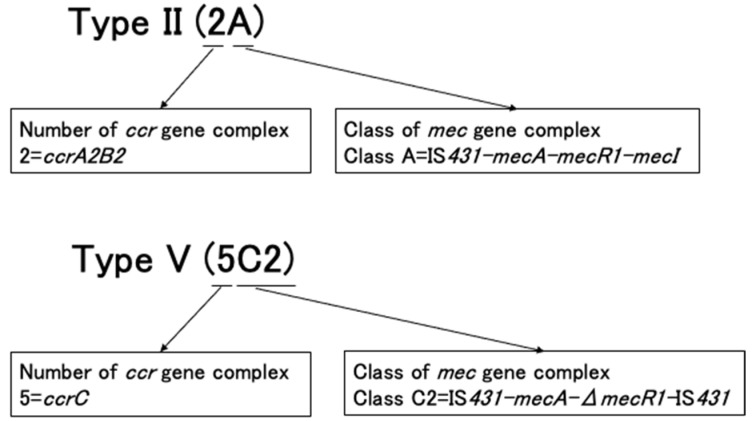
Description of SCC*mec* under two methods.

**Figure 5 antibiotics-11-00086-f005:**
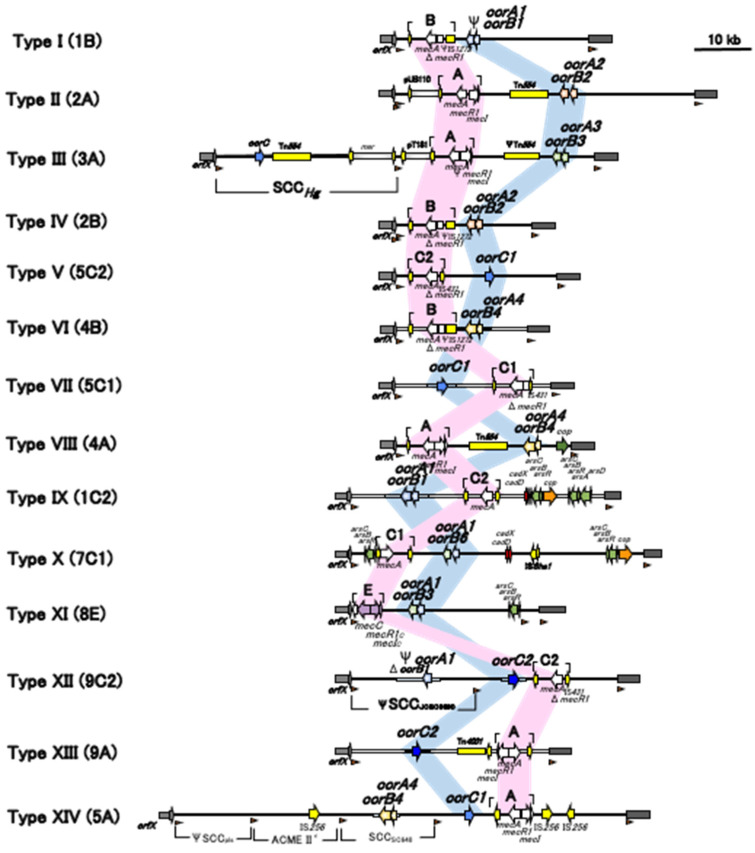
Schematic comparison of the current SCC*mec* types (14 December 2021). Adapted with permissions from Ref. [17]. 2009, American Society for Microbiology, and Ref. [28]. 2018, Dr. Marc Stegger, and Ref. [29]. 2020, Dr. Noriko Urushibara.

**Table 1 antibiotics-11-00086-t001:** List of *mecA* homologs shown in the IWG-SCC guidelines. Adapted with permission from Ref. [18]. 2012, American Society for Microbiology.

Prototype Strain	Reported *mec* Gene Name	New *mec* Gene Name	% Identity with the *mecA* Gene in *Staphylococcus aureus* N315
*Staphylococcus aureus* N315	*mecA*	*mecA*	100 (reference gene)
*Staphylococcal sciuri* K11	*mecA (mecA1)*	*mecA1*	79.1
*Staphylococcus vitulinus* CSBO8	*mecA*	*mecA2*	91
*Macrococcus caseolyticus* JCSC5402	*mecAm*	*mecB*	61.6
*Staphylococcus aureus* LGA251	*mecA* _LGA251_	*mecC*	68.7

**Table 2 antibiotics-11-00086-t002:** Current SCC*mec* types (14 December 2021).

SCC*mec* Type	*Ccr* Complex Type	*mec* Complex Class	Representative Strain (GenBank Accession No. or NCBI Reference Sequence No.)	Source	Country	Reported Time	Reference
I (1B)	1 (A1B1)	B	NCTC10442(AB033763)	human	United Kingdom, etc.	2001	[13]
II (2A)	2 (A2B2)	A	N315 (D86934)	human	Japan	2000, 2001	[12,13]
III (3A)	3 (A3B3)	A	85/2082 (AB037671)	human	New Zealand, etc.	2001	[13]
IV (2B)	2 (A2B2)	B	CA05(AB063172)	human	Japan	2002	[21]
V (5C2)	5 (C1)	C2	WIS(WBG8318) (AB121219)	human	Australia	2004	[22]
VI (4B)	4 (A4B4)	B	HDE288 (AF411935)	human	Portugal	2006	[23]
VII (5C1)	5 (C1)	C1	P5747/2002(AB373032)	human	Sweden	2008	[24]
VIII (4A)	4 (A4B4)	A	C10682 (FJ390057)	human	Canada	2009	[25]
IX (1C2)	1(A1B1)	C2	JCSC6943 (AB505628)	human (veterinarian)	Thailand	2011	[26]
X (7C1)	7(A1B6)	C1	JCSC6945 (AB505630)	human (veterinarian)	Canada	2011	[26]
XI (8E)	8(A1B3)	E	LGA251(FR821779, WGS)	bulk milk, daily cattle, human	England, Ireland, Denmark	2011	[19]
XII (9C2)	9(C2)	C2	BA01611 (KR187111)	cow	China	2015	[27]
XIII (9A)	9(C2)	A	55-99-44 (MG674089)	human	Denmark	2018	[28]
XIV (5A)	5 (C1)	A	SC792 (LC440647)	human	Japan	2019	[29]

**Table 3 antibiotics-11-00086-t003:** Current SCC*mec* I, II, IV and V subtypes (14 December 2021).

SCC*mec* Subtype	Representative Strain (GenBank Accession No. or NCBI Reference Sequence No.)	Source	Country	Reported Time	Subtyping by	Reference
SCC*mec* I subtype						
a	NCTC10442(AB033763)	human	United Kingdom, etc.	2001	(Reference)	[13]
b	PL72 (AB433542)	human	Poland	2006	J1	[31,32]
SCC*mec* II subtype						
a	N315 (NC_002745), Mu50 (NC_002758), MRSA252 (BX571856), JH1(NC_009632)	human	Japan	2001	(Reference)	[12,13]
b	JCSC3063(AB127982)	human	Japan	2005	J1	[33]
c	AR13.1/3330.2(AJ810120)	human	Ireland	2005	J1	[30]
d	RN7170 (AB261975: only J1 region)	human	United States (not clearly described)	2006	J1	[34]
e	JCSC6833(AB435013)	human	Japan	2009	J1	[31]
SCC*mec* IV subtype						
a	CA05(AB063172)	human	Japan	2002	(Reference)	[21]
b	8/6-3P (AB063173)	human	Japan	2002	J1	[21]
c	81/108 (AB096217)	human	Japan	2004	J1	[35]
d	JCSC4469 (AB097677)	human	Japan	2004	J1	[35]
g	M03-68 (DQ106887)	bovine milk	Korea	2005	J1	[36]
h	HO 5096 0412(EMRSA15) (HE681097)	human	Portugal, Greece, Finland	2007	J1	[37]
i	JCSC6668 (=CCUG41764)(AB425823)	human	Sweden	2009	J1	[38]
j	JCSC6670 (=CCUG27050) (AB425824)	human	Sweden	2009	J1	[38]
k	45394F(GU122149)	human (not clearly described)	Netherlands (not clearly described)	2010	J1	-
l	NN50 (AB633329)	human	Japan	2012	J1	[39]
m	JCSC8843 (AB872254)	human	Japan	2014	J1	[40]
n	No strain name found (KX385846.1)		Australia	2016	J3	-
o	No strain/accession number found in GenBank	human	Australia	2018	JI, J3	[41]
SCC*mec* V subtype						
a (5C2)	WIS(WBG8318) (AB121219)	human	Australia	2004	(Reference)	[22]
b (5C2&5)	TSGH17(=JCSC7190) (AB512767), PM1(AB462393), JCSC5952(AB478780)	human	Japan, Taiwan	2011	J1	[42]
c (5C2&5)	S0385(AM990992), JCSC6944(AB505629)	human (veterinarian)	(international pig conference)	2011	J1	[26]

## Data Availability

Not applicable.

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
