# Peer review of "Current Status of Staphylococcal Cassette Chromosome mec (SCCmec)"

_antibiotics, 2022, doi:10.3390/antibiotics11010086_

Round 1
Reviewer 1 Report
1. Resistance to methicillin is a worldwide distribution problem which occurs in
community-related infections as well as in nosocomial infections. The increase
in the frequency of infections caused by multi-resistant microorganisms makes
the study of distribution and diversity of genetic elements which confer
resistance to pharmaceuticals in bacteria of clinical importance necessary, as
well as the study of transmission mechanisms of such elements. It is good to
describe little more about methicillin in transmission elements.
2. It is clear that the diversity of elements in the S. aureus genome is really large, thus the best way to know this diversity would be through their sequencing. A novel assay for detection of methicillin-resistant Staphylococcus
aureus directly from clinical samples which has been recently published should
be stated.
3. Although being a crucial step in its horizontal transfer, little is known about the dynamics of SCCmec excision. The results presented in this review stated a
models in which excision occurs during a limited period of time at the early
stages of growth. The dynamics of the proportion of excitant's during growth s that excision occurred rather early on, at a given cell density, and during a
limited period of time should be more stated.
Author Response
Reply to Reviewer 1
I appreciate the critical but helpful review very much. I had prepared one-to-one response to each comment/recommendation and revised sentences were written in blue letters here in the reply.
- Resistance to methicillin is a worldwide distribution problem which occurs in community-related infections as well as in nosocomial infections. The increase in the frequency of infections caused by multi-resistant microorganisms makes the study of distribution and diversity of genetic elements which confer resistance to pharmaceuticals in bacteria of clinical importance necessary, as well as the study of transmission mechanisms of such elements. It is good to describe little more about methicillin in transmission elements.
Response 1: I appreciate the reviewer’s recommendation. It is difficult to know the direct role of methicillin for transmission of the elements, but wide distribution of MRSA is thought to be related with the use of methicillin and other antimicrobials. I added the sentences to describe the history of relatedness between prevalence of MRSA and antimicrobial use in the introduction part with some new references (P1, L29-37).
In the first report of MRSA, it was described that MRSA existed before starting use of methicillin but was rarely isolated in clinical settings, but after that, prevalence of MRSA in healthcare settings increased gradually. For example in the UK, the proportion of isolates of S. aureus from blood culture that were methicillin resistant increased in 1990s and reached approximately 40% around 2000, but decreased to less than 10% in 2010s [3,4]. High prevalence of antimicrobial resistance not only to methicillin but also to other antimicrobials was thought to have correlations with inappropriate use of antimicrobials, so the prevalence of MRSA is still used as an indicator of good infection control and prevention practices and appropriateness of antimicrobial usage [5,6].
- It is clear that the diversity of elements in the S. aureus genome is really large, thus the best way to know this diversity would be through their sequencing. A novel assay for detection of methicillin-resistant Staphylococcus aureus directly from clinical samples which has been recently published should be stated.
Response 2: I appreciate the reviewer’s recommendation. in the part of methods to identify SCCmec types and subtypes, I described the possibility in the future to use direct detection of MRSA from clinical samples combined with whole-genome sequencing to know diversity of SCCmec (P11, L284-294).
Recently, rapid method to detect MRSA, covering multiple SCCmec types and subtypes, directly from clinical samples has been developed [49]. Rapid confirmation of MRSA followed by whole-genome sequencing using same samples and implementation of SCCmecFinder may contribute to reveal the diversity of SCCmec in various settings. However, as mentioned later, objective confirmation of the structure is necessary to report new SCCmec, even though whole-genome sequencing has potential to discover new ones easily.
Although being a crucial step in its horizontal transfer, little is known about the dynamics of SCCmec excision. The results presented in this review stated a models in which excision occurs during a limited period of time at the early stages of growth. The dynamics of the proportion of excitant's during growth s that excision occurred rather early on, at a given cell density, and during a limited period of time should be more stated.
Response 3: I appreciate the reviewer’s recommendation. In the introduction part, I added the description of experimental excision and integration of SCCmec, and difficulty to integrate large SCCmec structures directly into S. aureus, with previous reference articles (P2, L60-73).
Precise excision of SCCmec and integration of plasmid containing ccr A and ccrB was proved by experiments within 24-hour incubation period using N315 having SCCmec II [12]. However, excision and integration of SCCmec from/to S. aureus isolate is thought to happen in complexed process in real clinical settings. It was observed that excision and integration of small SCCmec such as SCCmec I could happen, but integration of SCCmec II and V, which were larger than SCCmec I (>45 kb), could not happen even in vitro [15]. For large SCCmec structures, additional processes might be necessary to integrate into S. aureus, such as conjugative plasmids as a carrier, or subsequent multiple recombination events. As an example, it was reported that the two isolates from a patient with chronic S. aureus infection of intracardiac device changed from MSSA to MRSA during antimicrobial treatment, and the latter isolate possessed SCCmec II [16]. Some inversions and extra genetic fragment, which were not found in the original SCCmec II and its surrounding regions of N315, suggested that SCCmec II in the latter isolate did not directly integrated from N315 and additional processes might happen for integration.

Reviewer 2 Report
Authors fully analyzed the current status of SCCmec types, subtypes, and their different identification methods, as well as rules for nomenclature. It is a very interest review. However, some questions need to resolve before publishing:
Please use S. aureus in text, Figure and Table if it is not used firstly. And, please improve the quality of Figure 5 with higher resolution.
What does mean reference in Table 1? Please add it in notes.
“Reported in” in Tables 2 and 3 may be changed to Reported time.
In 5. Current SCCmec types and subtypes section, please simply describe Table 2 and Figure 5.
Please add the Conclusion section if it is possible.
In detection assays, they are weak and described simply. Please add the definitions, advantages and disadvantages of PCR, mPCR, genome sequencing-based methods, as well as references. Particularly, 6.3. Review of the methods may be combined into 6.1 and 6.2.
Other minors:
Please adjust the size of letter in Figure 4.
Non-standard format of References: such as Refs. 5, 6, 7, 21, 40, 41, etc.
Author Response
Reply to Reviewer 2
Authors fully analyzed the current status of SCCmec types, subtypes, and their different identification methods, as well as rules for nomenclature. It is a very interest review. However, some questions need to resolve before publishing:
I appreciate the critical but helpful review very much. I had prepared one-to-one response to each comment/recommendation and revised sentences were written in blue letters here in the reply.
The manuscript completed English language editing to make the description more accurate and natural. I added the comment about English language editing in the acknowledge part.
- Please use S. aureus in text, Figure and Table if it is not used firstly. And, please improve the quality of Figure 5 with higher resolution.
Response 1: I appreciate the reviewer’s recommendation. S. aureus is used throughout the manuscript including Tables and Figures after first description of S. aureus as an abbreviation of Staphylococcus aureus. I also noticed MSSA in Figure 1 should be fully spelled out because of its first appearance in the manuscript, so I modified the description of MSSA as “methicillin-susceptible S. aureus (MSSA)” in Figure 1. I prepared Figure 5 with higher resolution, but could not inoculate it in the format of manuscript. I would like to send the high-resolution figure to the editorial office independently.
- What does mean reference in Table 1? Please add it in notes.
Response 2: I appreciate the reviewer’s comment. Reference in Table 1 means “reference gene”. I revised the description (P4, L174).
- “Reported in” in Tables 2 and 3 may be changed to Reported time.
Response 3: I appreciate the reviewer’s comment. The description was changed to “reported time” according to the recommendation (P7, L224)
- In 5. Current SCCmec types and subtypes section, please simply describe Table 2 and Figure 5.
Response 3: I appreciate the reviewer’s recommendation. Detailed description about Table 2 and Figure 5 was added according to the recommendation (P6, L218-222).
Table 2 describes the details of the current SCCmec types, including the combinations of ccr complex and mec complex, representative strains and their source, country, and reported year. Figure 5 shows the comparison of the structures of SCCmec listed in Table 2.
- Please add the Conclusion section if it is possible.
Response 5: I appreciate the reviewer’s recommendation. I added the new part “Conclusion” at the end of manuscript, describing the history and future of SCCmec typing, usefulness of whole-genome sequencing and conventional PCR-based method, and continuous contribution of IWG-SCC (P12, L374-386).
- Conclusion
Discovery of SCCmec was based on the findings from multiple researches about MRSA in the 20th century, and its importance in the fields of molecular epidemiology, infection control and prevention, and investigation of evolution of MRSA, has been developed along with the evolution of molecular analysis technologies. Implementation of whole-genome sequencing will contribute to reveal the entire relatedness of SCCmec elements found not only in MRSA but also in other Staphylococcus species and Gram-positive bacteria. Importance as a tool of molecular typing might be declined if whole-genome sequencing is more widely used in clinical settings, but PCR-based methods will continue to be used as easy and costless tools for molecular epidemiology, infection control and prevention. Therefore, nomenclature rule of SCCmec should be followed, and IWG-SCC is expected to update the most appropriate policy to designate SCCmec elements, especially for isolates other than S. aureus, or MRSA isolated from animals or environment.
- In detection assays, they are weak and described simply. Please add the definitions, advantages and disadvantages of PCR, mPCR, genome sequencing-based methods, as well as references. Particularly, 6.3. Review of the methods may be combined into 6.1 and 6.2.
Response 6: I appreciate the reviewer’s recommendation. Review of the method in 6.3 mainly described multiplex PCR methods, so the section was merged to 6.1.. In 6.1, Definition of single PCR and multiplex PCR methods, and their advantages and disadvantages (P10 L252-260, L268-279).
Single PCR and multiplex PCR are used to identify SCCmec types. Single PCR for each genomic component in SCCmec is a basic, generally sensitive and specific method. SCCmec type is identified by the combination of detected components. However, it is difficult to guess SCCmec type before analysis and numerous different single PCRs are necessary for identification of SCCmec type. Therefore, multiplex PCR methods have been reported as useful tools to detect multiple components in SCCmec and identify SCCmec types and subtypes efficiently in one analysis, even though sensitivity and specificity to detect each component were sometimes less than single PCRs. The multiplex PCR methods to which IWG-SCC members contributed and were widely used are introduced here in this review.
The limitations of multiplex PCR methods should be considered when interpreting the results. For example, the partial deletion of the ccrB2 gene present in Japanese USA300, which was designated as ψUSA300, could cause failure in identifying the SCCmec type in some multiplex PCR methods [46]. In 2020, Dr. Yamaguchi published a substantially beneficial review covering the history and concept of SCCmec, and detailed methods of SCCmec typing using multiplex PCR formulated by Dr. Kondo [34,47]. In this review, limitation of multiplex PCR methods, such as problems to identify SCCmec type if one isolate had both of SCCmec and SCC (i.e. structure of SCCmec without mec complex) components, or failure of PCR caused by deletion/insertion of sequences of targeted sites. Approaches for solution of the problems were also mentioned, mainly about the necessity of additional single PCRs to detect indefinite components in SCCmec with different primer sets and PCR conditions.
In 6.2, detailed description about SCCmecFinder, advantages, possibility as well as concerns was added. (P10 L284-294, and description about the combination of rapid detection method of MRSA directly from clinical samples and whole-genome sequencing was added according to the comment from another reviewer).
SCCmecFinder identifies SCCmec elements in sequenced S. aureus isolates. Read data from major platforms for whole-genome sequencing, or preassembled genome/contigs can be submitted to the website of the SCCmecFinder. Users can obtain information about the prediction of SCCmec types based on genes in the ccr complex, mec complex, and J regions, including the homology to the entire cassette [48]. Recently, rapid method to detect MRSA, covering multiple SCCmec types and subtypes, directly from clinical samples has been developed [49]. Rapid confirmation of MRSA followed by whole-genome sequencing using same samples and implementation of SCCmecFinder may contribute to reveal the diversity of SCCmec in various settings. However, as mentioned later, objective confirmation of the structure is necessary to report new SCCmec, even though whole-genome sequencing has potential to discover new ones easily.
- Please adjust the size of letter in Figure 4.
Response 7: I appreciate the reviewer’s comment. The size of the letters in Figure 4 were adjusted as same size as Figure 3.
- Non-standard format of References: such as Refs. 5, 6, 7, 21, 40, 41, etc.
Response 8: I appreciate the reviewer’s comment. I added some references in the process of revision, so the description of the references was formatted again according to the author guide.
